Identification and expression analysis of the GDSL esterase/lipase family genes, and the characterization of SaGLIP8 in Sedum alfredii Hance under cadmium stress

Li He 1 2 3
Han Xiaojiao 2 3
Qiu Wenmin 2 3
Xu Dong 2 3
Wang Ying 2 3
Yu Miao 2 3
Hu Xianqi xqh@ynau.edu.cn xqhoo@126.com 1
Zhuo Renying zhuory@gmail.com 2 3
1 College of Plant Protection, Yunnan Agricultural University , Kunming , Yunnan , China
2 State Key Laboratory of Tree Genetics and Breeding, Chinese Academy of Forestry , Beijing , China
3 Key Laboratory of Tree Breeding of Zhejiang Province, The Research Institute of Subtropical of Forestry, Chinese Academy of Forestry , Hangzhou , Zhejiang , China
Uversky Vladimir
Electronic publication date: 2019 Apr 16
Publication date: 2019
Volume: 7
Electronic Location ID: e6741
Received 2019 Jan 25; Accepted 2019 Mar 7
Copyright: ©2019 Li et al.
Copyright year: 2019
Copyright holder: Li et al.
License: This is an open access article distributed under the terms of the Creative Commons Attribution License, which permits unrestricted use, distribution, reproduction and adaptation in any medium and for any purpose provided that it is properly attributed. For attribution, the original author(s), title, publication source (PeerJ) and either DOI or URL of the article must be cited.
License URL: https://creativecommons.org/licenses/by/4.0/

Keywords: Sedum alfredii Hance, GDSL esterase/lipase proteins (GELPs), Cadmium stress, SaGLIP8

Funding: National Nonprofit Institute Research Grant of Chinese Academy of Forestry CAFYBB2016SY008 National Natural Science Foundation of China 31872168 Zhejiang Science and Technology Major Program on Agricultural New Variety Breeding 2016C02056-1 Agricultural Projects of Public Scientific and Technology Research Zhejiang Province 2016C32G3030016 This work was supported by the National Nonprofit Institute Research Grant of Chinese Academy of Forestry (No. CAFYBB2016SY008), the National Natural Science Foundation of China (No. 31872168), the Zhejiang Science and Technology Major Program on Agricultural New Variety Breeding (No. 2016C02056-1) and the Agricultural Projects of Public Scientific and Technology Research Zhejiang Province (No. 2016C32G3030016). The funders had no role in study design, data collection and analysis, decision to publish, or preparation of the manuscript.

==============================
Background

The herb Sedum alfredii (S. alfredii) Hance is a hyperaccumulator of heavy metals (cadmium (Cd), zinc (Zn) and lead (Pb)); therefore, it could be a candidate plant for efficient phytoremediation. The GDSL esterase/lipase protein (GELP) family plays important roles in plant defense and growth. Although the GELP family members in a variety of plants have been cloned and analyzed, there are limited studies on the family’s responses to heavy metal-stress conditions.

Methods

Multiple sequence alignments and phylogenetic analyses were performed according to the criteria described. A WGCNA was used to construct co-expression regulatory networks. The roots of S. alfredii seedlings were treated with 100 µM CdCl2 for qRT-PCR to analyze expression levels in different tissues. SaGLIP8 was transformed into the Cd sensitive mutant strain yeast Δycf1 to investigate its role in resistance and accumulation to Cd.

Results

We analyzed GELP family members from genomic data of S. alfredii. A phylogenetic tree divided the 80 identified family members into three clades. The promoters of the 80 genes contained certain elements related to abiotic stress, such as TC-rich repeats (defense and stress responsiveness), heat shock elements (heat stress) and MYB-binding sites (drought-inducibility). In addition, 66 members had tissue-specific expression patterns and significant responses to Cd stress. In total, 13 hub genes were obtained, based on an existing S. alfredii transcriptome database, that control 459 edge genes, which were classified into five classes of functions in a co-expression subnetwork: cell wall and defense function, lipid and esterase, stress and tolerance, transport and transcription factor activity. Among the hub genes, Sa13F.102 (SaGLIP8), with a high expression level in all tissues, could increase Cd tolerance and accumulation in yeast when overexpressed.

Conclusion

Based on genomic data of S. alfredii, we conducted phylogenetic analyses, as well as conserved domain, motif and expression profiling of the GELP family under Cd-stress conditions. SaGLIP8 could increase Cd tolerance and accumulation in yeast. These results indicated the roles of GELPs in plant responses to heavy metal exposure and provides a theoretical basis for further studies of the SaGELP family’s functions.

Introduction

Cadmium (Cd) is an important environmental pollutant and inorganic toxicant, which has serious impacts on the growth and development of organisms (Liu et al., 2014). Cd has a wide range of sources, including electroplating, coatings and mining (Satarug et al., 2011), and can enter organisms through water and food, resulting in adverse effects (Nair et al., 2013). In mammals, Cd can cause a variety of diseases such as renal failure and blood pressure disorders, and it can also lead to osteoporosis, diabetes and neurological disorders (Jarup & Akesson, 2009; Messner & Bernhard, 2010). Cd accumulation in plants affects water balance and photosynthetic apparatus, resulting in leaf chlorosis, oxidative stress and stomatal opening inhibition (Oono et al., 2014). Heavy metal-contaminated soil threatens agriculture and food safety (Zhu et al., 2018). Some hyperaccumulative plants adapted to metalliferous soils in different ways (Gao et al., 2014). Hyperaccumulators, for instance, Cd/zinc (Zn)/lead (Pb) co-hyperaccumulator Sedum alfredii Hance, arsenic (As) and Pb co-hyperaccumulator Eremochloa ciliaris (Linn.) Merr. and manganese (Mn) hyperaccumulator Phytolacca acinosa Roxb are able to accumulate heavy metals in aboveground tissues but not exhibit symptoms of poisoning, and are widely used in phytoremediation (Pilon-Smits, 2005).

The hyperaccumulating ecotype S. alfredii Hance, with a high tolerance to Zn, Cd and Pb, can grow normally in soil having Cd concentrations up to 400 mg kg−1 (Tian et al., 2017; Xing et al., 2013). Leaf vacuolar isolation is currently considered to be the main mechanism of Cd detoxification in hyperaccumulator plants (Rascio & Navari-Izzo, 2011; Sharma, Dietz & Mimura, 2016). Genes related to Cd transport (Rascio & Navari-Izzo, 2011), chelation (Zhang et al., 2011) and reactive oxygen species (ROS) scavenging (Li et al., 2017) have been identified. SpHMA3 (Heavy metal ATPase 3) from Sedum plumbizincicola overexpressing in the non-hyperaccumulating ecotype of S. alfredii greatly increased its tolerance to, and cadmium detoxification is achieved by chelation of toxic or excessive heavy metals into the vacuole (Liu et al., 2017). In addition, SaCAX2 (cation exchanger 2), SaMT2 (metallothionein 2) and SaCu/Zn SOD (superoxide dismutase) isolated from S. alfredii in transgenic tobacco and Arabidopsis thaliana (A. thaliana) conferred greater tolerance levels to Cd stress (Liu et al., 2016; Zhang et al., 2014; Zhang et al., 2016). Overexpression of SaMT2 can chelate free cadmium in the cytoplasm and reduce the concentration of Cd. SaCAX2 can transport Cd into vesicles and store, however lipids that play important roles in abiotic stress remain largely unknown in this species. Signaling lipids can accumulate proteins on the membrane in an instant and affect the conformation and activity of proteins and metabolites in the cells, so that the plants can grow normally under abiotic stress conditions (Hou, Ufer & Bartels, 2016). Many lipase and esterase sequences have a pentapeptide GxSxG motif in which serine (S) is the central position of the conserved sequence. However, the hydrolysis/lipolytic enzyme subfamily GDSL has conserved motifs with different structures, the conserved amino acids are glycine (G), aspartic acid (D), S and leucine (L), and the active site serine is located near the N-terminus. The GDSL esterase/lipase protein (GELP) family has been identified in various plant species and is an attractive focus for scientists owing to their multifunctional nature in a wide range of organisms. Currently, there are 104 known GELP family members in Arabidopsis (Lai et al., 2017; Ling, 2008), and 130, 126, 96 and 57 family members in sorghum, Populus tomentosa, grape and Plutella xylostella, respectively (Volokita et al., 2011). GELPs are multifunctional hydrolytic enzymes that possess broad substrate specific and regiospecific activities. Consequently, the GELP enzymes are grouped in family II of the lipolytic enzymes (Akoh et al., 2004; Shakiba et al., 2016). They have four strictly conserved domains, I, II, III and V, which contain conserved Ser, Gly, Asn and His residues, respectively (Akoh et al., 2004; Molgaard, Kauppinen & Larsen, 2000). GELP family members have many functions in abiotic stress (Hong et al., 2008; Lee et al., 2009), morphogenesis (Ling et al., 2006), lipid metabolism (Brick et al., 1995) and seed development (Clauss et al., 2008; Riemann et al., 2007). AtGELP28 (SFAR2) and AtGELP59 (SFAR3) play key roles in plants under glucose-stress conditions (Chen et al., 2012). AtGELP60 (AtLTL1) enhances yeast tolerance to LiCl and might be involved in defense responses against pathogens (Naranjo et al., 2006). AtGLIP2 affects plant defense functions by inhibiting auxin responses (Lee et al., 2009).

However, S. alfredii GELP (SaGELP) gene has not yet been systematically identified under Cd stress or without Cd stress. Fortunately, the genome sequencing of S. alfredii has been completed by our group (R Zhuo, 2019, unpublished data), which enables the characterisation of the GELP family and their responses to Cd stress. In this study, we discovered 80 SaGELPs. A genome-wide bioinformatics analysis of the GELP family involved gene structures, phylogenetics and motif classification. In addition, the spatial–temporal expression patterns of SaGELP genes were determined under Cd-stress conditions. Finally, the heterologous expression of Sa13F.102 (SaGLIP8) in yeast increased Cd resistance and accumulation. These results provide the foundation for further studies on the functions of the GELP family, and the regulatory mechanisms of SaGELPs under heavy metal-stress conditions.

Material and Methods

Plant materials and stress treatments

Seedlings of the hyperaccumulator ecotype of S. alfredii were collected from an old Pb/Zn mining area in Quzhou City, Zhejiang Province, China. They were grown hydroponically in a growth chamber with day/night temperatures of 25 °C and a 16-h/8-h light/dark photoperiod. The seedlings were cultured in 1/2−strength Hoagland’s solution for 4 weeks. Subsequently, the roots of the experimental seedlings were immersed in 100 µM CdCl2 as the stress treatment, while the seedlings of the control group were further cultured in 1/2-strength Hoagland’s solution. Roots, stems and leaves were sampled at 0 h, 6 h and 7 d. Three biological repeats per sample were taken at each time point and stored in a −80 °C refrigerator for subsequent use.

Identification of SaGELPs in S. alfredii Hance

The OrthoMCL algorithm (Li, Stoeckert Jr & Roos, 2003) was used to analyze 16 species, S. alfredii and 15 other related species, Phalaenopsis equestris, Oryza sativa, A. thaliana, Populus trichocarpa, Amborella trichopoda, Rhodiola crenulata, Kalanchoe fedtschenkoi, Medicago truncatula, Brassica rapa, Solanum lycopersicum, Daucus carota, Coffea canephora, Nelumbo nucifera, Macleaya cordata and Ananas comosus. The HMMER search (https://www.ebi.ac.uk/Tools/hmmer/) (Finn, Clements & Eddy, 2011) was conducted to identify and screen for possible SaGELPs containing Lipase_GDSL (Pfam: PF00657) domains in S. alfredii (Chen et al., 2018). The GELP homologous sequences in A. thaliana were obtained from TAIR (https://www.arabidopsis.org/index.jsp). All of the candidate SaGELPs were further confirmed by SMART (http://smart.embl-heidelberg.de/) (Letunic & Bork, 2018) according to AtGELPs. The basic information for SaGELPs were predicted using ExPASy (https://web.expasy.org/protparam/) (Mariethoz et al., 2018), including molecular weights, amino acid numbers and isoelectric point values.

Multiple sequence alignments and phylogenetic analyses

All of the validated SaGELP and selected AtGELP protein sequences were aligned with ClustalX in MEGA5 using GONNET as the protein weight matrix, with a gap opening penalty of 10 and gap extension penalty of 0.1. Phylogenetic trees were constructed using the Neighbor-joining method with the following parameters: text of phylogeny = bootstrap method; number of bootstrap replications = 1,000; and gaps/missing data treatment = complete deletion. iTOL (http://itol.embl.de/upload.cgi) tools were used to modify the phylogenetic trees (Letunic & Bork, 2016).

Gene structure and conserved motif predictions

Gene structure diagrams of SaGELPs were obtained from Gene Structure Display Server 2.0 (http://gsds.cbi.pku.edu.cn/) (Hu et al., 2014). Conserved motifs in SaGELPs sequences were identified using Multiple Expectation Maximization for Motif Elicitation (MEME, e value < 1e−10) services (http://meme-suite.org/tools/meme) (Bailey et al., 2015), with the following parameters: motif discovery = classic mode; number of repetitions = 0 or 1 occurrence per sequence; maximum number of motifs = 50; and optimum motif width = 6–100 residues. The consensus blocks in conserved domains were constructed using WebLogo (http://weblogo.berkeley.edu/logo.cgi) (Crooks et al., 2004).

Analysis of cis-regulatory elements from promoters

The cis-regulatory elements in the promoters were predicted in the 1.5 kb upstream regions of all SaGELP genes using the online website PlantCARE (http://bioinformatics.psb.ugent.be/webtools/plantcare/html/) (Lescot et al., 2002). Different cis-responsive elements in the promoters were presented using RAST (http://rsat.eead.csic.es/plants/feature-map_form.cgi).

Co-expression network construction

A weighted gene co-expression network analysis (WGCNA) was used to construct co-expression regulatory networks based on profiles of differentially expressed gene responses to Cd stress, as described by Han et al. (2016). The Pearson’s correlation coefficient of the Fragments Per Kilobase of transcript per Million fragments mapped (FPKM) value of each gene pair was calculated using the R programming language, with the correlation coefficient threshold set to 0.30 (Han et al., 2016). We screened the members of the SaGELP family and identified hub genes in the co-expression network (Langfelder, Mischel & Horvath, 2013). All eligible edges were classified according to their annotations, and we further analyze their associations with hub genes (Lotia et al., 2013). Finally, the co-expression subnetwork was visualized with Cytoscape v3.6.1 (Shannon et al., 2003).

Total RNA isolation and expression analysis

Total RNA of S. alfredii treated with 100 µM CdCl2 was extracted from all roots, stems and leaves, using an RNA extraction kit (NORGEN, Thorold, ON, Canada). RNase-free DNaseI (New England BioLabs, Ipswich, MA, USA) was used to process genomic DNA and digest all samples. PrimeScript™ RT Master Mix (TaKaRa, Dalian, China) (Stephens, Hutchins & Dauphin, 2010) was used to produce the first-strand cDNA, which was stored at −80 °C for later use.

Quantitative Real-Time PCR (qRT-PCR) reactions were carried out using the SYBR® Green premix Ex Taq™ (TaKaRa) reagent on the thermal circulator of an Applied Biosystems 7300 Real-Time PCR System (Applied Biosystems, Foster City, CA, USA) (Chen et al., 2018). Sequences of primers used in qRT-PCR are shown in Table S1. The relative expression level of each SaGELP gene was calculated based on the comparison threshold period (2−ΔΔCT) method, using SaUBC9 as an endogenous reference gene (Sang et al., 2013). The heat map of the relative expression levels was constructed using online software at OmicShare (http://www.omicshare.com/). The qRT-PCR products of the expected size were analyzed by 1.5% agarose gel electrophoresis.

Heterologous expression of SaGLIP8 in yeast

The specific primers SaGLIP8-F/R were used to amplify the open reading frame of SaGLIP8 (Table S1). The purified PCR product was first inserted into the entry vector pDONR222 (Invitrogen, Carlsbad, CA, USA), and then yeast expression vector pYES-DEST52-SaGLIP8 were constructed by gateway LR reaction. The empty vector pYES2.0 was used as a control. Two vectors, expression vector pYES-DEST52-SaGLIP8 as well as empty vector pYES2.0, were transformed into the Cd sensitive mutant strain Saccharomyces cerevisiae (Δycf1) using the lithium acetate method (Liu et al., 2016). Positive colony selection was performed in the solid medium with 50 µg ml−1 ampicillin and PCR reaction. The selected positive clones in the yeast liquid were cultured to an OD600 value of 0.8–1.0, and then spotted on SG-U (synthetic galactose-uracil) solid medium containing concentrations of 0, 15 and 30 µM CdCl2. The strains in SG-U liquid medium were diluted (OD600 = 100, 1/10, 1/100, 1/1000, 1/10000 and 1/100000), then incubated in a 28 °C incubator for 3 d (Chen et al., 2017; Liu et al., 2016). In addition, two transformed yeast cell strains were cultured on liquid SG-U medium containing 30 µM CdCl2 for 96 h at 28 °C to determine the Cd accumulation levels by the Inductively Coupled Plasma Mass Spectrometry (ICP-MS, NexIon 300D, Perkin Elmer, Shelton, CT, USA).

Results

Eighty SaGELP family members were identified and classified into three clades

A total of 80 SaGELPs and 56 pseudoenzymes (incomplete domain structure) were dug out (Table S2). All of the characteristics of the 80 SaGELP candidate genes are listed in Table 1, including the amino acid lengths, molecular weights and theoretical isoelectric point values. The coding sequence (CDS) lengths ranged from 900 bp (Sa28F.37) to 1,920 bp (Sa9F.272), with an average length of 1,131 bp. In total, 104 candidate sequences were obtained from A. thaliana through preliminary research, and the HMM analysis confirmed 101 AtGELP sequences and 4 pseudoenzymes (Table S3). We further used AtGELPs and SaGELPs to build the phylogenetic tree. The SaGELP gene family is divided into clades I, II and III, and the numbers of subclades are 13, 6 and 2, respectively (Fig. 1), consistent with a previous study of terrestrial plant AtGELPs (Volokita et al., 2011), showing that three branches, two major and one minor, existed in its phylogenetic tree.

Table 1 Analysis of amino acid sequence information of Sedum alfredii Hance GELP family.

Name	GeneBank accession no.	CDS length (bp)	Number of AA	Molecular weight	Theoretical pI	
Sa0F.11	MK440731	1,089	362	40,634.89	8.38	
Sa0F.223	MK440732	1,143	380	41,846.12	9.22	
Sa0F.262	MK440733	1,056	351	39,364.84	6.45	
Sa0F.41	MK440734	1,089	362	40,526.79	8.46	
Sa0F.898	MK440735	1,113	370	42,075.78	4.93	
Sa105F.31	MK440736	1,095	364	40,161.02	9.24	
Sa10F.217	MK440737	1,104	367	40,867.46	5.07	
Sa10F.441	MK440738	1,056	351	39,493.17	5.91	
Sa110F.22	MK440739	1,197	398	44,141.56	5.19	
Sa116F.110	MK440740	1,155	384	42,046.70	5.80	
Sa116F.65	MK440741	1,320	439	49,170.05	8.67	
Sa121F.27	MK440742	1,086	361	39,895.98	8.51	
Sa129F.2	MK440743	1,092	363	39,815.06	8.87	
Sa12F.49	MK440744	1,092	363	38,720.53	5.16	
Sa136F.34	MK440745	1,134	377	41,922.28	8.34	
Sa13F.102	MK440746	1,098	365	40,683.49	9.54	
Sa13F.118	MK440747	1,125	374	41,088.11	8.51	
Sa14F.252.1	MK440748	1,188	395	44,124.62	6.04	
Sa184F.22	MK440749	1,089	362	40,195.00	5.28	
Sa18F.151	MK440750	1,164	387	42,818.94	8.86	
Sa1F.117	MK440751	1,032	343	37,924.76	9.22	
Sa207F.22	MK440752	1,119	372	41,258.41	8.65	
Sa20F.120	MK440753	1,359	452	50,719.60	9.00	
Sa217F.43	MK440754	999	332	36,974.89	5.43	
Sa248F.34	MK440755	1,062	353	37,905.99	6.64	
Sa24F.223	MK440756	1,059	352	38,951.99	5.69	
Sa24F.262	MK440757	1,089	362	39,028.77	8.59	
Sa258F.41	MK440758	1,095	364	40,739.54	5.32	
Sa26F.146	MK440759	1,131	376	40,913.88	7.52	
Sa26F.158	MK440760	1,092	363	40,043.83	5.06	
Sa27F.42	MK440761	1,149	382	42,465.01	6.25	
Sa28F.36	MK440762	1,119	372	40,698.35	5.37	
Sa28F.37	MK440763	900	299	32,786.50	5.95	
Sa28F.38	MK440764	1,098	365	40,515.96	4.98	
Sa28F.39	MK440765	1,095	364	39,581.04	6.02	
Sa29F.188.1	MK440766	1,146	381	40,894.65	8.72	
Sa29F.343	MK440767	1,068	355	40,295.08	4.84	
Sa2F.358	MK440768	1,053	350	38,328.89	9.14	
Sa314F.6	MK440769	1,113	370	40,383.85	8.65	
Sa32F.165	MK440770	1,092	363	39,855.27	6.98	
Sa33F.119	MK440771	1,179	392	43,638.93	6.24	
Sa36F.86	MK440772	1,116	371	40,462.25	6.74	
Sa39F.196	MK440773	1,149	382	42,338.07	8.44	
Sa39F.291	MK440774	1,206	401	44,639.75	5.03	
Sa3F.277	MK440775	1,104	367	39,972.74	8.30	
Sa3F.554	MK440776	1,107	368	41,055.21	6.26	
Sa42F.134	MK440777	1,107	368	40,177.26	8.73	
Sa45F.55	MK440778	1,074	357	39,953.75	9.41	
Sa46F.118	MK440779	1,146	381	43,149.62	9.44	
Sa46F.20	MK440780	1,095	364	40,835.13	8.38	
Sa46F.270	MK440781	1,131	376	41,403.07	8.31	
Sa46F.31	MK440782	1,089	362	40,230.55	9.20	
Sa47F.286	MK440783	1,140	379	42,340.79	5.50	
Sa55F.146	MK440784	1,167	388	42,873.82	7.05	
Sa56F.143	MK440785	1,104	367	40,895.18	8.48	
Sa57F.156	MK440786	1,137	378	41,578.62	5.45	
Sa5F.25	MK440787	1,056	351	38,963.46	8.38	
Sa5F.658	MK440788	1,089	362	40,100.03	8.81	
Sa64F.90	MK440789	1,107	368	40,530.32	5.98	
Sa66F.83	MK440790	1,188	395	43,742.49	5.59	
Sa6F.163.1	MK440791	1,197	398	44,232.55	5.11	
Sa6F.233	MK440792	1,089	362	39,775.35	8.72	
Sa72F.20	MK440793	1,068	355	38,596.68	8.71	
Sa79F.141	MK440794	1,164	387	42,555.48	5.52	
Sa79F.142	MK440795	1,122	373	40,791.61	5.86	
Sa79F.143	MK440796	1,131	376	41,701.89	8.07	
Sa7F.179	MK440797	1,071	356	39,256.81	8.39	
Sa7F.184	MK440798	1,107	368	40,817.82	5.55	
Sa7F.458	MK440799	1,104	367	40,610.49	9.19	
Sa7F.504	MK440800	1,143	380	42,515.73	9.40	
Sa81F.175	MK440801	1,107	368	40,903.24	9.06	
Sa82F.57	MK440802	1,074	357	38,294.41	6.58	
Sa84F.200	MK440803	1,140	379	41,043.23	5.00	
Sa86F.120	MK440804	1,098	365	40,085.52	7.55	
Sa92F.24	MK440805	1,128	375	42,017.14	5.68	
Sa95F.131	MK440806	1,119	372	41,139.90	5.33	
Sa99F.11	MK440807	1,113	370	40,647.49	6.54	
Sa9F.272	MK440808	1,920	639	61,736.39	9.03	
Sa9F.31	MK440809	1,164	387	42,896.80	8.41	
Sa9F.81	MK440810	1,557	518	53,135.19	8.59	
Notes.

CDS Coding sequence

AA amino acid

pI isoelectric point

Figure 1 Phylogenetic relationships among the S. alfredii and Arabidopsis thaliana GELP families.

The tree was generated with ClustalW and MEGA 5.0 software using the Neighbor-joining method. SaGELPs are labeled in red, and AtGELPs are labeled in black. Different clades are represented by different colored backgrounds. Different sub-branches are represented by different colored branching lines. The pale blue solid circle represents leaf sorting. The size of circle corresponds to the bootstrap value.

Gene structure and conserved motif analysis

The genomic sequences ranged from 1,226 bp (Sa24F.223) to 3,312 bp (Sa79F.143) (Table S4). The two genes (Sa116F.65 and Sa99F.11) had maximum number of exons (seven), while the three genes (Sa39F.196, Sa9F.272 and Sa9F.81) contained only two exons. The gene structure analysis showed that the average number of exons was five, within 56 SaGELP genes (70%) containing five exons and four introns (Fig. 2B, Table S5).

Figure 2 Gene structures and motif compositions of the GELP family in S. alfredii.

(A) The phylogenetic tree of SaGELP amino acid sequences was inferred using the Neighbor-joining method and 1000 bootstrap replicates; (B) Gene structures of SaGELPs. Yellow boxes represent CDS, blue boxes represent upstream or downstream, lines represent introns; (C) Schematic representation of each of the conserved motifs in the SaGELPs selected by the MEME online tool. Different motifs are represented by different colored boxes.

Conserved sequences and motifs represent important sites for enzymatic functions. Among the 23 discovered motifs (Fig. 2C, Table S6), we analyzed four conservative motifs in the SaGELPs, blocks I, II, III and V (2, 5, 7 and 1, respectively), and the different blocks contained different motifs (Fig. 3, Table S7). A total of 13 well-conserved motifs were found (E values < 1e−100) in most SaGELP genes, while other motifs were specific to individual SaGELPs (Table S7). Motifs 14 and 18 were only found in clade I, while motifs 16 and 21 were only discovered in clade II. Motif 23 was unique in clades I and II, and motifs 20 and 22 were distributed in clades I and III. The others motifs existed in all clades (Fig. 1A).

Figure 3 Four conserved motifs of S. alfredii GELPs.

The four consensus blocks I, II, III and V represent the conserved motifs in the amino acid sequences of the GDSL family, and are numbered based on their location from the N to C terminal. Triangles represent the conserved catalytic residues in conserved domains. Ser (S), Gly (G), Asn (N) and His (H) are conserved residues.

Analysis of cis -regulatory elements from promoters

The 1.5 kb upstream regulatory regions of the SaGELP genes were explored for stress-related regulatory elements (Table S8). The cis-acting element analysis of all genes is shown in Fig. 4. We identified cis-regulatory elements related to hormones, such as auxin, gibberellin, methyl jasmonate and ethylene. TC-rich (ATTTTCTCCA) repeats are related to cis-acting elements involving in defense and stress response. Meanwhile, some elements are also related to abiotic stress, such as heat shock elements (heat stress) and MYB-binding (AACCTAA, MRE) sites (drought-inducibility).

Figure 4 Cis-acting element analysis of promoters from the GELP family in S. alfredii.

Different elements are represented by different colored boxes. The box size corresponds to the element’s sequence length.

Co-expression network of SaGELPs

A large number of hub genes regulate potential target genes, including those related to general tolerance mechanisms and responses to Cd stress. Here, a total of 13 hub genes related to SaGELPs were obtained, as well as potential edge genes. The co-expression regulatory network involved 13 hub genes and 5 regulated different functional groups from GO (Gene Ontology) annotation (Table S7). Most of the co-expressed genes are involved in metabolic processes, growth and development, catalytic activity and biological regulation, indicating that SaGELPs have multiple functions in plants. We selected the edge genes involved in Cd tolerance from several regulatory networks, including cell wall and defense function, lipid and esterase, stress and tolerance, transport and transcription factor activity (Table S7).

As shown in Fig. 5, the major categories were transport (254 edges), transcription factor (112 edges), lipid and esterase (63 edges), cell wall and defense function (24 edges), and stress and tolerance (6 edges). The hub gene Sa0F.898 had the largest module in the Cd response gene co-expression network, with 128 nodes, including 60, 29, 19, 7 and 3 nodes related to transport, transcription factor, lipid and esterase, cell wall and defense function, and stress and tolerance, respectively. Other hub genes were also associated with different biological functions. For example, Sa13F.102 was mainly related to lipid, esterase, cell wall and defense function, while Sa26F.146 was mainly involved transport function. Therefore, SaGELPs might be involved in the induction of stress signals and function by activating transcription factors to regulate genes involved in metal transport. In addition, they was related to the enhancement of the plant’s resistance to heavy metals.

Figure 5 Co-expression network of S. alfredii GELP genes.

Nodes indicate genes, and edges indicate significant co-expression events between genes. Target genes involved in the same process are grouped together, and different groups are distinguished by different colors.

Tissue expression patterns and Cd response profiles

We used qRT-PCR to understand the functions of the SaGELP genes in S. alfredii and the tissue expression pattern under Cd-stress conditions at three time points (0 h, 6 h and 7 d). The tissues expression profiles of the genes were converted into a heat map on the basis of their expression levels (Fig. 6). All of the expression levels of SaGELP genes could be divided into the following three cases: (1) 75 genes expression significantly up-regulated at 6 h and decreased at 7 d in root (such as Sa28F.36, Sa5F.25 and Sa46F.20), stem (such as Sa27F.42, Sa45F.55 and Sa314F.6) and leaf (such as Sa13F.118, Sa0F.41 and Sa42F.134); (2) 7 up-regulated expression in stems and leaves (such as Sa10F.217); and (3) 12 down-regulated trends in roots, stems and leaves (such as Sa105F.31, Sa7F.458 and Sa184F.22).

Figure 6 Expression profiles of S. alfredii GELPs in root (R), stem (S) and leaf (L) under normal and cadmium (Cd)-stress conditions.

The heat map shows the expression of 80 SaGELP genes. Each small square represents a gene, and its color represents the expression of the gene. The greater the expression, the darker the color (red, up-regulated; green, down-regulated). The “0” represents the control without Cd stress. The stress time were six hours (6 h) and seven days (7 d).

Furthermore, the gene expression levels were greatly different in untreated samples (without Cd treatment). We performed a data analysis on the hub genes in roots, stems and leaves. Sa13F.102, Sa28F.36 and Sa5F.25 were constitutively expressed at relatively high levels in root (Fig. 7A), while the three most highly expressed SaGELPs in the stem were Sa13F.102, Sa5F.25 and Sa29F.188.1 (Fig. 7B), and the three most highly expressed SaGELPs in the leaf were Sa13F.102, Sa95F.131 and Sa29F.188.1 (Fig. 7C). Sa13F.102 had the highest expression level in all tissues. Meanwhile, the results from qRT-PCR gel image were in accordance with gene expression levels (Fig. S2).

Figure 7 The expression profiles of 13 S. alfredii GELP hub genes in different tissues under normal conditions.

(A) root; (B) stem; (C) leaf. The expression level of the control, Sa26F.146, (y-axis “Relative mRNA expression”) was arbitrarily set to 1. Bars indicate means ± standard deviations (SDs) of at least three independent biological replicates.

SaGLIP8 heterologous expression enhanced Cd tolerance and accumulation in yeast

Due to the recent relationship between the Sa13F.102 gene and At5G45670.1 (AtGLIP8) from the above phylogenetic tree (Fig. S1), we designated Sa13F.102 gene as SaGLIP8. As a hub gene in the co-expression network, Sa13F.102 (SaGLIP8) was selected for functional verification for its strong induction in response to Cd stress, which implied vital roles in the Cd response in all three tissues. SaGLIP8 gene was expressed in the Cd sensitive mutant strain Saccharomyces cerevisiae (Δycf1). The SaGLIP8-overexpressive yeast grew better than the pYES2.0 yeast on a medium containing 15 and 30 µM CdCl2, suggesting that the SaGLIP8 gene could increase Cd tolerance in yeast (Fig. 8A). Cd concentration measurements, with pYES2.0 as the control, revealed that the Cd content of SaGLIP8-overexpressive yeast was significantly greater than that of pYES2.0, and the difference between the two was extremely significant (P = 0.01) (Fig. 8B).

Figure 8 Overexpression of S. alfredii Hance GLIP8 increases the cadmium (Cd) tolerance and accumulation in yeast.

(A) the growth of Δycf1 yeast mutants transformed with the empty vector pYES2.0 or with pYES-DEST52 harboring SaGLIP8; (B) the accumulation of Cd in Δycf1 yeast cells. Bars indicate means ± standard deviations (SDs) of at least three independent biological replicates. Two asterisks indicate a significant difference at p < 0.01 compared with an empty vector.

Discussion

Plants can accumulate such things as heavy metals, due to environmental pollution, through activating the expression of corresponding proteins involved in stress response, including phytochelatins (PCs) and metallothioneins (MTs) (Hasan et al., 2017). Hyperaccumulator plants accumulate heavy metals in the body and exhibit enhanced tolerance levels; consequently, they can be used for phytoremediation and other purposes (DalCorso, Manara & Furini, 2013; Tian et al., 2016). S. alfredii is a hyper-accumulation plant, which absorbs and accumulates Cd from the soil. Thus, it is a promising candidate plant species to alleviate and solve soil pollution problems (Clemens et al., 2013). There have been many reports on Cd absorption and dynamic balance in S. alfredii (Liu et al., 2016; Tian et al., 2017); however, the molecular mechanism underlying Cd detoxification in S. alfredii remains poorly understood.

GELP family members have been reported in many plant species, and have many roles, including in abiotic stress responses and defense functions (Abdelkafi et al., 2009; Cao et al., 2018; Dong et al., 2016; Lai et al., 2017; Tan et al., 2014). GLIP1 in pepper can participate in wound defense responses (Hong et al., 2008). OsGLIP1 and OsGLIP2 proteins from O. sativa are located in lipid droplets and endoplasmic reticulum membranes and play a key role in lipid metabolism and immune response (Gao & Yin, 2017). The GELP family plays important roles in plant abiotic stress responses, but research on their roles under Cd-stress conditions was limited. In this study, 80 SaGELP genes of S. alfredii were identified. A phylogenetic analysis showed that SaGELPs could be divided into three main groups (Fig. 1), which was consistent with the classifications reported by previous researchers (Lai et al., 2017; Ling et al., 2006). The responses of the GDSL family to biotic and abiotic stresses had been studied (Hong et al., 2008; Shakiba et al., 2016), but there were few studies on the effects of heavy metal contamination. Therefore, it was necessary to investigate the effects of heavy metals on the SaGELP family of genes.

In Brassica napus L., the BnLIP2 gene was expressed in a tissue-specific manner and was abundantly expressed during seed germination (Ling et al., 2006). Was the GELP family also expressed in a tissue-specific manner in S. alfredii? To answer this question, we treated roots of S. alfredii seedlings with 100 µM CdCl2 and used three different time points to construct a heat map to observe expression changes. The roots, stems and leaves under 0 h were used the qRT-PCR. We then analyzed the results and used the online software to draw the heat map. Relative expression values were calculated by Z-score normalization. Most of the genes showed significant changes, which were related to S. alfredii’s ability to co-excessively accumulate Cd (Fig. 6). The hub gene Sa13F.102 was abundantly expressed in all tissues (roots, stems and leaves) under normal conditions (0 h), and other hub genes were abundantly expressed in specific tissues, such as Sa28F.36’s expression in roots and Sa5F.25’s expression in stems (Fig. 7). Thus, some SaGELP gene family members were expressed in a tissue-specific manner. After Cd treatments, some genes also showed specificity of expression, such as Sa28F.36 and Sa28F.38, which were only up-regulated in roots, Sa10F.217 and Sa29F.343, which were only up-regulated in stems, and Sa0F.41 and Sa12F.49, which were only up-regulated in leaves. As members of an esterase or lipase gene family (Akoh et al., 2004), some SaGELP genes might be associated with cell wall synthesis (Zhang et al., 2017) and stress (Shakiba et al., 2016). Consequently, we selected 13 hub genes that had edge genes with these or related functions according to their gene ontology classification and constructed a co-expression regulatory network (Fig. 5). Most of the identified genes were involved in transport (254 edges) and a few were associated with stress responses (6 edges).

We predicted that SaGLIP8 encodes an extracellular protein. As shown in Fig. 8, SaGLIP8 could increase the Cd tolerance and content in transgenic yeast. Thus, we hypothesize that this protein may function like the OsGLIP1 and OsGLIP2 proteins. Some reported GEIP genes can be regulated in a variety of ways to enhance their defense functions. Brittle leaf sheath1 (BS1) in rice is a member of the GELP family and is involved in the formation of this cell wall and plays an important role in the defense function of plants (Zhang et al., 2017). The GELP genes, especially SaGLIP8 in S. alfredii, can also be regulated by several means, which may improve Cd tolerance.

Conclusions

Based on genomic data of S. alfredii, we conducted phylogenetic analyses, as well as conserved domain, motif and expression profiling of the GELP family under Cd-stress conditions. The phylogenetic trees were constructed by combining the A. thaliana and S. alfredii GELP family genes, which indicated that the associated domains were conserved during evolution. According to the structural and phylogenetic characteristics of the SaGELP sequences, they were divided into three clades. Most of the genes were responsive to Cd stress. In total, 13 hub genes were obtained, and a co-expression regulatory subnetwork was constructed. The edge genes mainly had five functions. Additionally, SaGLIP8 (Sa13F.102) was cloned into an expression vector and transformed into yeast. SaGLIP8 enhanced Cd tolerance and accumulation in yeast. This result indicated the roles of GELPs in plant responses to heavy metal exposure and provides a theoretical basis for further studies of the SaGELP family’s functions.

Supplemental Information

Table S1 qRT-PCR primer sequences

qRT-PCR, Quantitative Real-Time PCR. The primer sequences were designed by primer 5.

Click here for additional data file.

Table S2 Conding sequences and amino acid of all genes

The direction of conding sequences is from 5′to 3′. Sa, Sedum alfredii.

Click here for additional data file.

Table S3 The sequences of AtGELPs

The sequences represents amino acid of AtGELPs. AT, Arabidopsis thaliana.

Click here for additional data file.

Supplemental Information 4 Genomic sequences of SaGELPs

The direction of genomic sequences is from 5′to 3′. Sa, Sedum alfredii.

Click here for additional data file.

Table S5 The number of exons and introns in SaGELPs

Blue represents the intron. CDS, coding sequence. Sa, Sedum alfredii.

Click here for additional data file.

Table S6 Conserved motifs identified in the SaGLIP proteins

The pictures of motif were downloaded from online website MEME ( http://meme-suite.org/tools/meme). Sa, Sedum alfredii.

Click here for additional data file.

Table S7 The hub genes and edge genes

GO, Gene Ontology. KEGG, Kyoto Encyclopedia of Genes and Genome.

Click here for additional data file.

Table S7 The promoter sequences of SaGELPs

The direction of promoter sequences is from 5′to 3′. Sa, Sedum alfredii.

Click here for additional data file.

Figure S1 Phylogenetic relationship of S. alfredii Hance 13F.102 and the AtGELP gene family

The tree was generated with ClustalW and MEGA 5.0 software using the Neighbor-joining method. The number of bootstrap replications is 1,000.

Click here for additional data file.

Supplemental Information 10 The qRT-PCR gel image of 13 S. alfredii hub genes in different tissues under normal conditions

(A) root; (B) stem; (C) leaf. UBC9 is reference gene of S. alfredii. M represents 2,000 bp DNA ladder marker. Relative expression values were calculated by Z-score normalization. Green and red showed the low and high expression levels, respectively. The names of the samples are exhibited at the bottom.

Click here for additional data file.

Additional Information and Declarations

Competing Interests

Author Contributions

Data Availability

The authors declare there are no competing interests.

He Li conceived and designed the experiments, performed the experiments, analyzed the data, prepared figures and/or tables, authored or reviewed drafts of the paper.

Xiaojiao Han conceived and designed the experiments, analyzed the data, prepared figures and/or tables, authored or reviewed drafts of the paper, approved the final draft, genome analysis.

Wenmin Qiu analyzed the data, authored or reviewed drafts of the paper.

Dong Xu and Miao Yu helped with the use of analysis software.

Ying Wang performed the experiments.

Xianqi Hu conceived and designed the experiments, authored or reviewed drafts of the paper, approved the final draft.

Renying Zhuo conceived and designed the experiments, contributed reagents/materials/analysis tools, authored or reviewed drafts of the paper, approved the final draft.

The following information was supplied regarding data availability:

The raw measurements are provided in Tables S1–S4, S8.

Data is available at GenBank, accession numbers Genbank:MK440731 to Genbank:MK440810.

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
