# Peer review of "Identification and expression analysis of the GDSL esterase/lipase family genes, and the characterization of SaGLIP8 in Sedum alfredii Hance under cadmium stress"

_PeerJ, doi:10.7717/peerj.6741_

## Round 0.1 · original submission · Minor Revisions

As you can see, all reviewers have found your manuscript interesting and important. However, they also raised several minor concerns. Please carefully address all these concerns and revise your manuscript accordingly.

Reviewer 1 ·

Basic reporting

see general comments for the author

Experimental design

see general comments for the author

Validity of the findings

see general comments for the author

Additional comments

The manuscript is well developed and clearly structured, and the conclusions are supported by the results. I only have a few comments.
1. For the title, I suggest to change " Identification and expression analysis of the GDSL
esterase/lipase family" to "Identification and expression analysis of the GDSL
esterase/lipase family genes", otherwise, it sounds like the the manuscript is about identification and purification of GDSL esterase/lipase proteins, not genes.
2. What does GDSL stand for?
3. The English language needs to be improved. For example, Line 29, I suggest change "experimental materials" to roots of S.alfredii seedlings. Line 31 and 32, "were" should be was, "investigate the resistance and accumulation to Cd" change to "investigate its role in resistance and accumulation to Cd". Line 59 "are adapted to" should be "adapt to". Line 243 "For the..." change to "due to the..".
4. The genes studied are described as both GDSL gene/family/enzyme and GELP gene/family/enzyme, are GDSL and GELP equivalent?
5 Line 292, "As shown in Figure 6, SaGLIP8 is also an inhibitory protein". Figure 6 only describes the gene expression level, hence, it is not clear to me why SaGLIP8 is an inhibitory protein, does SaGLIP8 inhibit the function of any other protein? Please rephrase this sentence and add more explanation.
6. Line 293 and 294, "Some reported GLIP genes can be regulated in a variety of ways to enhance their defense functions" needs a reference.

Reviewer 2 ·

Basic reporting

In this manuscript, the authors have performed extensive genomic analysis and identified 80 genes that belong to GDSL esterase/lipase protein family in S. alfredii. Furthermore, authors have identified the gene, SaGLIP8, involved in the cadmium detoxification. The manuscript is well-written and provides sufficient background for readers.

Experimental design

The methods are properly described. This study will be of interest to researchers working in this specific field. Furthermore, the authors provide all the raw data files with their analysis and mentioned all the tools and databases used for the study.

Validity of the findings

No comment

Additional comments

1. line 74: the number of GELP in Arabidopsis is written 108 whereas in line 180: 105 known sequences in Arabidopsis is mentioned.
2. line 195: Table S7 is mentioned. Is it correct?
3. line 254-255: the sentence seems to be incomplete. What are the "corresponding proteins"? Mention the name of the proteins.
4. In the result section: Gene structure and conserved motif analysis": The authors should show the conserved motifs representation in the three-dimensional structure.
5. In Fig. 7, the authors show the bar graphs based on heat map as shown in Fig. 6. The authors identified some specific genes highly expressed in roots, stems and leaves. It is better to show the RT-PCR gel image of these specific genes also.

Reviewer 3 ·

Basic reporting

This is a very well-written manuscript showcasing preliminary data of the function of SaGLIP8 gene under cadmium stress.
Some comments are as follows:
Abstract:
1. abbreviations are used without defining them first.
2. under "methods" in abstract, line 29 - "...experimental materials...." - please mention what these are.

Introduction:
1. Lines 52 - 59 - Authors introduce Cd as a pollutant that has serious impacts on growth and development. Then talk about sources of Cd as pollutant and methods of accumulation in organism followed by what the harmful effects are. Re-arrange these sentences such that impact on growth and development is immediately followed by examples of those impacts.
2. Lines 59-60 - what are some examples of how plants tolerate high metals?
3. Lines 60-61- examples of hyperaccumulators
3. Lines 67, 69 - some genes are mentioned that improve tolerance for Cd. However, what are their basic functions? Is it known? Please define what these genes are.
4. Line 71 - what are lipid proteins? It seems to be a vague term. Does it imply proteins involved in lipid metabolism or proteins that contain fatty acid modifications? How does the lipid part of the organism tie into studying heavy metal stress?
5. Line 73 - Please define GDSL before using the abbreviation.Also please introduce what the GDSL family is before discussing GELP family proteins.
6. Line 79-81 - When discussing the structure of the protein, please have a figure, either molecular or schematic to refer to. How does the protein structure tie in with the motifs that have been discussed later in the discussion section of the manuscript? This is not discussed or mentioned anywhere. A discussion would make the study more complete.
7. Line 87 - Authors say "....has not yet been systematically identified under Cd stress..." Doe this mean that the gene has been studied when there is no heavy metal stress?
8. Line 90 - Please define SaGELPs before using the abbreviation.
9. Line 90 - Authors say ..."obtained..." Should it instead read "discovered" instead? As the authors are the one who are analysing the genome?

Experimental design

Good experimental design.

Line 133 - spelling mistake. "promoters" instead of "promotres"
Line 149 - in the description of this method, relate back to "Plant materials and stress treatments" as that is where the roots, stems and leaves are from.

Validity of the findings

The results and discussion section is well written. Some comments are as follows:

Results section:

Line 235 - What does "untreated" mean? Please state clearly.
Line 252 - Is there a numerical value to the "significant difference"?

Discussion section:

Line 266 - Please examine the sentence and make sure it is complete.
Line 276 - Please define what "samples" mean.
Line 276 -277 - please discuss shortly how the heat map was obtained.
Line 291-292 - How does figure 6 show that SaGLIP8 is a inhibitory protein?

---

## Round 0.2 · accepted · Accept

Since all the critical points of all reviewers were adequately addressed and since the manuscript was revised accordingly, I am pleased to accept this manuscript in its present form.

# LINE NO: / BEFORE / AFTER / [COMMENTS]
LINE 42: / overexressed. / overexpressed. / [.]
LINE 75: / and store. However, lipids / and store; however, lipids / [reads better, but not perfect due to short incomplete sentence.]
LINE 266: / to environmental pollution such as / due to environmental pollution such things as / [Suggested, reads better.]
LINE 267: / involving in stress response, such as / involved in stress response, including / [.]
LINE 289: / In this experiment / To answer this question / [Better applied since previous sentence is a question.]
LINE 291: / Then analyze the results, / We then analyzed the results and / [.]
#